# Phenotypic Diversity Analysis in *Elaeagnus angustifolia* Populations in Gansu Province, China

Rongrong Shi [1], Zhu Zhu [1,2,*], Ningrui Shi [1], Yongmei Li [1], Jun Dang [1], Yanli Wang [1], Yonglong Ma [1], Xiangyun Xu [1] and Ting Liu [1]

[1] College of Forestry, Gansu Agricultural University, Lanzhou 730070, China; shirr1204@163.com (R.S.)
[2] Wolfberry Harmless Cultivation Engineering Research Center of Gansu Province, Lanzhou 730070, China
* Correspondence: zhuz@gsau.edu.cn; Tel.: +86-13-321-317-121

**Abstract:** As a highly resistant urban ornamental plant, *Elaeagnus angustifolia* L. is often used in dry land, saline-alkali land shelter forest, and landscape horticulture. It is the main windbreak and sand-fixing tree species in Gansu Province, China. The special geographical and climatic environment makes the distribution and growth of *E. angustifolia* in Gansu Province show different degrees of difference. In order to evaluate the phenotypic diversity of *E. angustifolia* in different populations and its variation patterns under different geographical and climatic conditions, 35 phenotypic traits (trunk, branch, leaf, and flower related traits) of 90 plants from 10 populations in Gansu Province were measured and analyzed. The results showed the following: (1) *E. angustifolia* has rich phenotypic variation. The variation is greater among populations. The traits with the largest and smallest coefficients of variation were "under-branch height" and "flower diameter", respectively. The variation in the Qilihe population was the largest, and the variation in the Ganzhou population was the smallest. The diversity of flowers and leaves is relatively higher. (2) Correlation analysis showed that most of traits were closely related. Leaf traits showed a gradient variation law dominated by altitude and precipitation. Flower traits were affected by the synergistic effects of various geographical and climatic factors. (3) The results of the principal component analysis (PCA) showed that the primary traits affecting the phenotypic diversity of *E. angustifolia* were leaf size and branch length among the related traits of trunk, branch, leaf, and flower. (4) Cluster analysis showed that 90 *E. angustifolia* plants were clustered into four clusters that were not completely clustered according to geographical distance and may be randomly affected by genotypic or environmental factors. These results will lay a foundation for further analysis of the genetic mechanism of phenotypic traits of *E. angustifolia* and also provide a reference for the collection, preservation, and variety improvement of *E. angustifolia* germplasm resources.

**Keywords:** *Elaeagnus angustifolia*; phenotypic diversity; geographical climate variation; multivariate analysis

## 1. Introduction

Phenotypic traits are the most intuitive manifestations of plant growth showing diversity and are affected by their own genetic characteristics and environmental factors [1]. Plant growth reflects the adaptation of genotypes to environmental changes, irreversible changes in long-term stress selection, and new phenotypes after stable inheritance. Therefore, phenotypic variation is often crucial in adaptation and evolution [2]. Phenotypic diversity is an important indicator and main research variable used to measure species diversity. Studying the phenotypic diversity of plants and exploring the differences among species or populations not only helps in understanding the response of plants to the ecological environment, and then exploring their development and evolution [3–5], but also contributes to the protection and management of germplasm resources [6], which is of great

significance for the introduction and breeding of plants. At the same time, the study of phenotypic traits is also used to define new varieties to be protected. Because the information generated by plant morphological characterization comes from large datasets composed of qualitative and quantitative features, the use of multivariate analysis is particularly suitable for this situation [7]. Among them, correlation analysis, principal component analysis, and cluster analysis are the most common and effective analysis methods in phenotypic diversity research. At present, they have been applied to the phenotypic analysis of many plants, such as *Hyphaene compressa*, *Crataegus orientalis*, *Citrus microcarpa*, etc. [8–10].

Gansu Province is located in the western region of China. It is located at the intersection of the upper reaches of the Yellow River and the Loess Plateau, the Qinghai–Tibet Plateau, and the Inner Mongolia Plateau. The topography of Gansu Province is narrow and long, and the landform and climate types are complex and diverse [11]. From southeast to northwest, it extends into a narrow zone of 1600 km. The mountains, plateaus, plains, valleys, deserts, and Gobi are interlaced, and the geographical environment is complex. The climate type spans three thermal zones: subtropical, warm temperate, and temperate zones. The precipitation decreases from more than 600 nm to less than 100 mm in the northwest. The regional differences are obvious, and the biodiversity is also regionally distributed, which makes the distribution and growth of plants show different degrees of difference [12,13]. The genetic diversity of the fruit phenotypic traits of natural populations of *Calligonum mongolicum* in Gansu Province was studied by Tian et al. It was found that the phenotypic variation degree of 12 natural populations of *C. mongolicum* was higher, and the phenotypic variation showed a gradient regularity with latitude and longitude variation [14]. Luo et al. studied the phenotypic characteristics and diversity of walnut nuts in Gansu Province. It was found that the walnut resources in this region had rich phenotypic diversity, and the variation among populations was small. The variation within populations was the main source of phenotypic diversity of walnut nuts in Gansu Province. The results of cluster analysis were positively correlated with geographical distance or climatic conditions [15].

*Elaeagnus angustifolia* L. is a deciduous tree or shrub of *Elaeagnus*, which is native to southern Europe and Asia [16]. In China, it is mainly distributed in the northwest provinces and western Inner Mongolia. *E. angustifolia* has the characteristics of wind and sand resistance, salt tolerance, drought resistance, high-temperature resistance, barren resistance, and easy reproduction [17]. It has been a popular shrub for windbreaks and shelterbelts in semi-arid and saline environments because of its adaptability [16]. *E. angustifolia* also has high ornamental value. It has a unique tree shape, color, and fragrance. Its highly esthetic appearance and life forms favor its use in landscape horticulture (amenity parks and forests, hedges, the greening of streets, and public spaces, etc.) [18]. It is also an economic tree species with multiple functions. Its branches, leaves, flowers, and fruits have the values of development and utilization. At present, it is widely used for food, medicine, papermaking, foraging, wood, and furniture in China. *E. angustifolia* plays a great role in ensuring stable agricultural harvests, ecological environment maintenance, and landscape construction in Gansu and other places.

As a multifunctional tree species with high ecological, ornamental, and economic value, scholars from various countries have conducted a wide range of research on *E. angustifolia*. These studies have mainly focused on its pharmacological effects [19], edible value [20], active ingredient extraction [21], and stress resistance [22], etc., and research on the phenotypic traits of *E. angustifolia* is relatively lacking. In previous studies, Safdari et al. used multivariate analysis to study its phenotypic diversity related to fruit quality in Iran [23]. Zeng et al. conducted a phenotypic diversity analysis and ornamental evaluation of *E. angustifolia* in Zhangye City, Gansu Province [24]. Xu et al. [25] and Fan et al. [26] analyzed the diversity of germplasm resources of *E. angustifolia* in Gansu, Inner Mongolia, Ningxia, and other places. However, there is a lack of in-depth research on the comprehensive phenotypic analysis of the plant shape, branches, leaves, and flowers of *E. angustifolia* and its correlation with environmental factors, which is not conducive to the further ex-

ploration of the growth law and ornamental value of *E. angustifolia*. From long-term field observations, we found that *E. angustifolia* in Gansu Province has formed a rich resource diversity under a complex geography, climate, and long cultivation history, and different regions have significant phenotypic differences. Therefore, some reasons for the phenotypic diversity of *E. angustifolia* can be explained by exploring its phenotypic variation, so as to understand the response of *E. angustifolia* growth to the ecological environment and also provide some basis for the genetic improvement, diversity conservation, and resource management of *E. angustifolia* [27].

In this study, 10 populations of *E. angustifolia* in Gansu Province were used as the research object, and 35 phenotypic traits related to the plant shape, branch, leaf, and flower of *E. angustifolia* were measured and analyzed. Our main objectives are the following: (1) to estimate whether the phenotypic variation mainly comes from within the population or among populations; (2) to understand the variation degree and diversity level of *E. angustifolia* phenotypes in different environments; and (3) to reveal the relationship between phenotypic variation and geographical and climatic factors. Our research results are of great significance to the establishment of the identification and evaluation system of *E. angustifolia* germplasm resources and will lay a data foundation for the introduction, development, and directional breeding of high-quality varieties.

## 2. Materials and Methods

### 2.1. Plant Material

According to the distribution records of *E. angustifolia* in the Chinese Virtual Herbarium (http://www.cvh.ac.cn/, accessed on 12 August 2022) and the field investigation, this study selected 10 counties as the research population sites in the areas where *E. angustifolia* is distributed in Gansu Province. The *E. angustifolia* in the study area is mostly distributed in the suburbs, roads, or villages, and a small part is distributed in parks or temples. Almost all of these areas, there is soil drought and poor site conditions. In the field investigation, a GPS positioning system was used to obtain the altitude, longitude, and latitude of each group. The climate data were the cumulative average values of 10 county/district meteorological stations in the surrounding area of the sampling points published by the National Meteorological Data of China (http://data.cma.cn/, accessed on 12 August 2022) from 1981 to 2010. The detailed information is shown in Figure 1 and Table 1.

**Table 1.** Geographical and climatic information and sample sizes of 10 populations.

| Population | Sample Size | Altitude (m, AL) | Longitude (°E, E) | Latitude (°N, N) | Annual Mean Temperature (°C, AMT) | Annual Precipitation (mm, AP) | Annual Mean Relative Humidity (%, AMRH) |
|---|---|---|---|---|---|---|---|
| Dunhuang (DH) | 11 | 1168.2 | 94.66 | 40.14 | 9.9 | 42.2 | 40 |
| Suzhou (SZ) | 10 | 1492.0 | 98.51 | 39.75 | 7.8 | 88.4 | 48 |
| Linze (LZ) | 10 | 1453.1 | 100.26 | 39.09 | 8.3 | 113.4 | 49 |
| Ganzhou (GZ)) | 5 | 1515.3 | 100.38 | 39.00 | 7.8 | 132.6 | 52 |
| Yongchang (YC) | 5 | 1728.7 | 102.06 | 38.37 | 5.4 | 211.8 | 52 |
| Minqin (MQ) | 20 | 1481.7 | 103.15 | 38.59 | 8.8 | 113.2 | 44 |
| Gulang (GL) | 10 | 1792.4 | 102.97 | 37.60 | 5.7 | 352.3 | 51 |
| Qilihe (QLH) | 7 | 1544.5 | 103.74 | 36.07 | 10.5 | 360.0 | 60 |
| Yongjing (YJ) | 3 | 1967.3 | 103.39 | 35.98 | 9.7 | 273.7 | 59 |
| Linxia (LX) | 9 | 2025.1 | 103.19 | 35.61 | 7.3 | 501.3 | 67 |

Because of the COVID-19 pandemic, the survey and sampling time of this study were performed from 25 May 2022 to 5 June 2022. In each population, 3–20 plants with normal growth, no obvious defects and diseases and insect pests, and a plant age > 5a were randomly selected as research samples (in principle, the number of samples in each population should have been more than 10, but in some populations, due to the lack of

samples or sampling difficulties, the number of samples is less than 10), and the spacing of samples was more than 15 m. The plant height, diameter at breast height, and branch height of the sample plants were measured, respectively. At the same time, 10 groups of complete branches, leaves, and florets were evenly selected from the upper, middle, and lower parts of each sample plant, and the relevant indicators were measured on site.

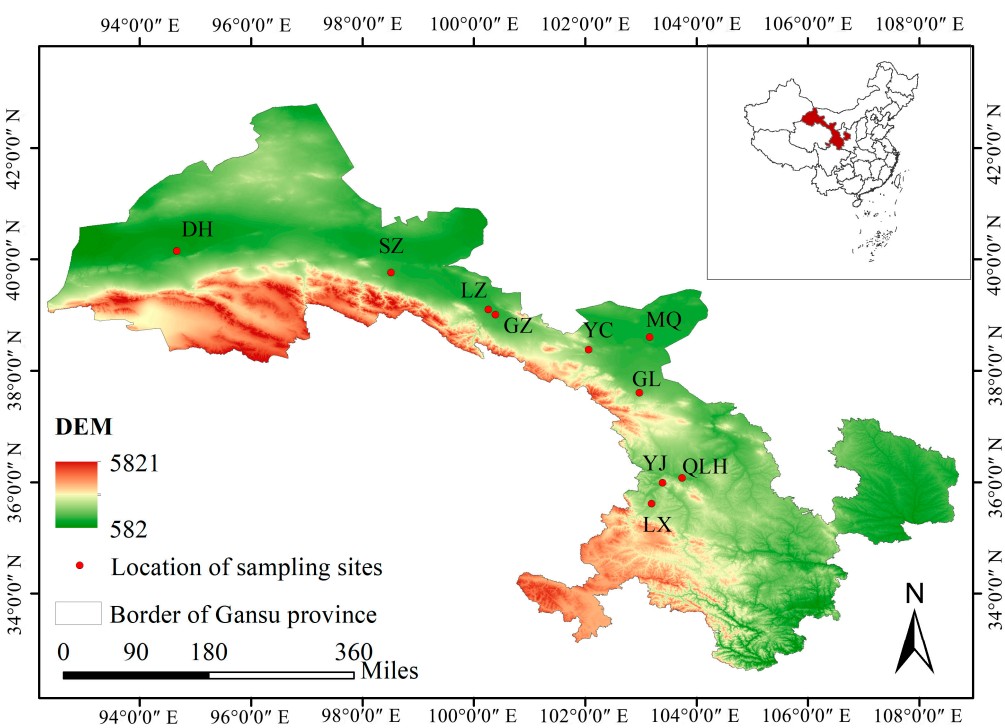

**Figure 1.** Geographical distribution of various populations of *E. angustifolia*. Geo-elevation data were derived from an SRTM DEMUTM 90 M resolution digital elevation in the Geospatial Data Cloud (http://www.gscloud.cn/, accessed on 12 August 2022). ArcGIS 10.7 geographic information analysis software was used for data processing to generate distribution maps.

*2.2. Morphometric Analysis*

The phenotypic traits of *E. angustifolia* were divided into quantitative traits and quality traits. Quantitative traits, including 21 traits such as tree height, crown diameter, and diameter at breast height, were quantitatively evaluated (Table 2). The qualitative traits included 14 traits such as trunk type, crown roundness, canopy density, and the qualitative evaluation method was used (Table 3).

**Table 2.** Quantitative traits and abbreviations of *E. angustifolia*.

| No. | Character | Abbreviation | Unit |
|:---:|:---:|:---:|:---:|
| 1 | Tree height | TH | m |
| 2 | Crown diameter | CrD | cm |
| 3 | Diameter at breast height | DBH | cm |
| 4 | Under-branch height | UBH | cm |
| 5 | Branching angle | BA | ° |
| 6 | Branching number | BN | – |
| 7 | Leaf length | LL | mm |
| 8 | Leaf width | LW | mm |
| 9 | Leaf thickness | LTh | mm |
| 10 | Petiole length | PL | mm |
| 11 | Leaf shape index | LSI | – |
| 12 | Leaf area | LA | mm$^2$ |
| 13 | Annual branch length | ABL | mm |

**Table 2.** *Cont*.

| No. | Character | Abbreviation | Unit |
|---|---|---|---|
| 14 | Number of flower clusters on annual branches | NFC | – |
| 15 | Ratio of flower cluster number to branch length | RFB | – |
| 16 | Flower diameter | FD | mm |
| 17 | Calyx tube length | CTL | mm |
| 18 | Calyx tube width | CTW | mm |
| 19 | Flower stalk length | FSL | mm |
| 20 | Flower color number | FCN | – |
| 21 | Floret number in axils of leaves | FNAL | – |

LSI = LL/LW, RFB = NFC/ABL. The "Flower color number (FCN)" is the total number of different flower colors on the same tree.

**Table 3.** Grading assignment of quality traits of *E. angustifolia*.

| No. | Character | Abbreviation | Grading Assignment | | | | | | |
|---|---|---|---|---|---|---|---|---|---|
| | | | 1 | 2 | 3 | 4 | 5 | 6 | 7 |
| 1 | Trunk type | TT | Multi-trunk shrubs | Less trunk shrubs | Arbor | – | – | – | – |
| 2 | Crown roundness | CR | Messy | Moderate | Rounded | – | – | – | – |
| 3 | Canopy density | CaDe | Low | Medium | High | – | – | – | – |
| 4 | Tree growth vigor | TGV | Low | Medium | High | – | – | – | – |
| 5 | Branch thorn | BT | Present | Absent | – | – | – | – | – |
| 6 | Branch color | BC | Hay yellow | Brownish green | Red brown | Dark brown | – | – | – |
| 7 | Leaf shape | LS | Oval-shaped | Long oval | Ovate | Lanceolate | Narrow lanceolate | – | – |
| 8 | Leaf apex shape | LAS | Obtuse | Acuminate | – | – | – | – | – |
| 9 | Speckles on leaves | SL | Low | Medium | High | – | – | – | – |
| 10 | Leaf upper surface color | LUSC | Aloe gray(a) * | Aloe gray(b) | Aloe gray(c) | Aloe gray(d) | Dark olive green | Grass green(a) | Grass green(b) |
| 11 | Leaf lower surface color | LLSC | Aloe gray(a) | Aloe gray(b) | Aloe gray(c) | – | – | – | – |
| 12 | Petal spreading state | PSS | Explanate | Curling outward | – | – | – | – | – |
| 13 | Flower density degree | FDD | Low | Medium | High | – | – | – | – |
| 14 | Flower color | FC | Rice white | Beige | Pale yellow | Bright yellow | Orange | – | – |

\* The lowercase letters represent different shades of the same color. The level a is the lightest, and the color is deepened from a to d.

Tree height was measured by a tree-height-measuring instrument. The crown diameter, diameter at breast height, and under-branch height were measured by a tape measure. The branching number (based on the first branch on the ground), number of flower clusters on annual branches, flower color number, and floret number in the axils of leaves were counted and recorded. Annual branch length, leaf length, leaf width, petiole length, leaf thickness, flower diameter, calyx tube length, calyx tube width, and flower stalk length were measured by a scale and vernier caliper. Leaf area was measured by a leaf area analyzer (WinRhizo; Regent Instruments Inc., Quebec, QC, Canada). Flower color, leaf

upper surface color, leaf lower surface color, and branch color were compared and recorded with color cards (Figure 2).

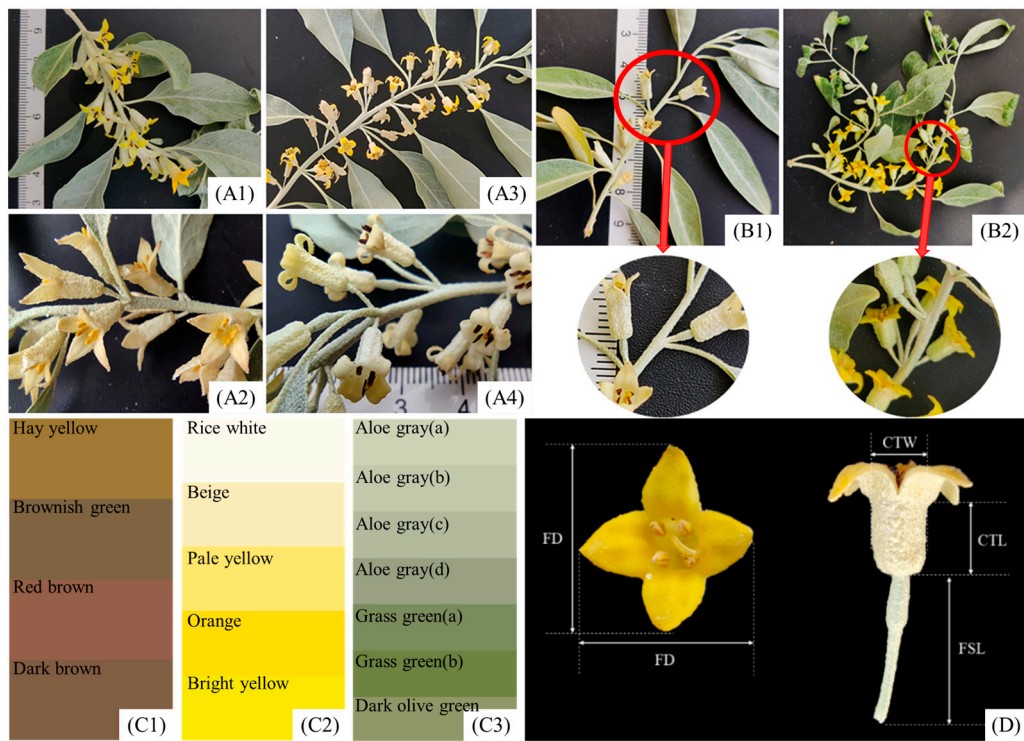

**Figure 2.** Legend of some traits of *E. angustifolia*. (**A**) PSS—Petal spreading state: explanate (**A1**,**A2**); curling outward (**A3**,**A4**). (**B**) FNAL—Floret number in axils of leaves: (**B1**) 1 floret; (**B2**) 3 florets. (**C**) The color of each part of *E. angustifolia*. (**C1**) Branch color (**C2**) Flower color (**C3**) Leaf color (**D**) Measurement of floral traits: FD—Flower diameter; CTL—Calyx tube length; CTW—Calyx tube width; FSL—Flower stalk length.

### 2.3. Statistical Analysis

#### 2.3.1. Analysis of Variance

Statistically significant differences within and among populations were determined by nested analysis of variance. The linear model comprises the following:

$$Y_{ijk} = \mu + \tau_i + \delta_{j(i)} + \varepsilon_{ijk}$$

$Y_{ijk}$ is the *k*th observed value of the *j*th family in the *i*th population, $\mu$ is the total average, $\tau_i$ is the effect value among populations, $\delta_{j(i)}$ is the random effect value of the individual plant (family) in the population, and $\varepsilon_{ijk}$ is the random error [28].

#### 2.3.2. Phenotypic Differentiation Coefficient

The phenotypic differentiation within and among populations was reflected by the phenotypic differentiation coefficient ($V_{ST}$). The formula comprises the following:

$$V_{ST} = \frac{\sigma_{t/s}^2}{\sigma_{t/s}^2 + \sigma_s^2}$$

$\sigma_{t/s}^2$ is the variance component among populations, and $\sigma_s^2$ is the variance component within the population (family) [27,29].

### 2.3.3. Descriptive Statistics

Descriptive statistics were performed on the morphometrics of each population using SPSS 20.0 software, including the arithmetic mean (X), standard deviation (SD), Duncan's multiple comparison, and coefficient of variation (CV) to determine the range of variation [30].

$$CV = SD/\bar{X}$$

### 2.3.4. Diversity Index

The diversity index of each trait was calculated by EXCEL2010 software. The Shannon–Wiener index (H′) is a quantitative expression used to describe the degree of variation of trait diversity. The calculation formula is the following:

$$H' = -\Sigma(P_i \times lnP_i)$$

H′ is the diversity index and $P_i$ is the effective percentage of the distribution frequency in the $i$th grade material of a trait. The distribution frequency of quality traits is expressed as the percentage of the number of sample plants at a certain level of traits to the total number of sample plants. The average value and standard deviation of the quantitative traits were calculated, and the obtained quantitative trait data were graded and standardized. According to the total average (X) and standard deviation (SD), they were divided into 10 levels. Among them, level 1 was less than $\bar{X} - 2SD$, level 10 was greater than $\bar{X} + 2SD$, and the difference between each level was 0.5SD [24,31].

### 2.3.5. Multivariate Analysis

Multivariate statistical methods were used to further determine the structure and differences in the phenotypic traits of *E. angustifolia*. The correlation between each trait and the correlation between traits and geographical climatic factors were obtained by Pearson's correlation analysis. Principal component analysis (PCA) was used to reduce the dimension of each index to obtain the key indexes affecting the phenotype of *E. angustifolia*. Loading plot and biplot analysis were performed according to the obtained PC1, PC2, and PC3. Ward cluster analysis was used to divide the studied samples into different groups, and a circular tree diagram was constructed with the square Euclidean distance as the measurement method [32]. Before the clustering analysis, the original data were processed by Z standardization (STD) to eliminate the influence of different dimensions on the analysis. The above statistical analysis and graphics were completed by SPSS20.0 and Origin 2022 software.

## 3. Results

### 3.1. Phenotypic Differences among and within Populations

The variance analysis of *E. angustifolia* phenotypic traits showed that 29 traits were significantly different among populations ($p < 0.01$), and only nine traits were significantly different within populations ($p < 0.01$). It can be seen that most phenotypic traits were widely different among different populations, but there was no significant difference among different plants in the same population (Table 4).

According to the results of the nested variance analysis, the variance components and phenotypic differentiation coefficients of 35 phenotypic traits of *E. angustifolia* among and within populations were calculated, respectively (Table 4). The phenotypic differentiation coefficients of each trait among populations ranged from 21.29% to 92.59%, with an average of 54.15%. The traits with the largest and smallest phenotypic differentiation coefficient were FNAL and FDD. This shows that the floret number in axils of leaves is more differentiated among populations, while the flower density degree is less differentiated and relatively stable. The average variance components of phenotypic traits among and within populations accounted for 27.28% and 21.49% of the total variation, respectively, indicating

that there was a certain degree of variation in the phenotypic traits of *E. angustifolia* among and within populations, and the variation was greater among populations.

**Table 4.** Variance analysis and phenotypic differentiation coefficients among the *E. angustifolia* populations.

| Traits | F Value | | Proportion of Variance Components (%) | | | Phenotypic Differentiation Coefficients (%) |
|---|---|---|---|---|---|---|
| | Among Populations | Within Populations | Among Populations | Within Populations | Random Errors | |
| TH | 4.125 ** | 1.919 ** | 27.58 | 27.09 | 45.33 | 50.45 |
| CrD | 2.458 ** | 0.8 | 22.50 | 15.46 | 62.04 | 59.28 |
| DBH | 4.075 ** | 1.949 ** | 32.76 | 24.37 | 42.87 | 57.34 |
| UBH | 3.076 ** | 0.682 | 32.48 | 11.21 | 56.31 | 74.34 |
| BA | 0.996 | 1.217 | 12.11 | 23.02 | 64.86 | 34.47 |
| BN | 0.503 | 1.117 | 6.64 | 22.95 | 70.41 | 22.45 |
| LL | 3.319 ** | 1.751 | 24.06 | 26.80 | 49.14 | 47.31 |
| LW | 4.416 ** | 2.048 ** | 28.45 | 27.87 | 43.68 | 50.52 |
| LTh | 4.527 ** | 1.22 | 32.50 | 18.75 | 48.75 | 63.41 |
| PL | 2.992 ** | 1.16 | 24.49 | 20.04 | 55.47 | 54.99 |
| LSI | 2.793 ** | 1.432 | 22.17 | 24.01 | 53.82 | 48.01 |
| LA | 4.824 ** | 1.636 | 32.04 | 22.95 | 45.02 | 58.27 |
| ABL | 7.656 ** | 1.416 | 43.94 | 17.16 | 38.90 | 71.92 |
| NFC | 3.849 ** | 2.122 ** | 25.48 | 29.65 | 44.87 | 46.22 |
| RFB | 2.779 ** | 2.394 ** | 19.02 | 34.59 | 46.39 | 35.48 |
| FD | 4.658 ** | 0.963 | 34.59 | 15.09 | 50.32 | 69.62 |
| CTL | 2.574 ** | 0.776 | 23.42 | 14.91 | 61.67 | 61.10 |
| CTW | 17.57 ** | 4.657 ** | 51.41 | 28.76 | 19.83 | 64.12 |
| FSL | 5.911 ** | 1.254 | 38.55 | 17.26 | 44.20 | 69.07 |
| FCN | 2.031 | 0.595 | 20.18 | 12.47 | 67.35 | 61.80 |
| FNAL | 14.212 ** | 0.539 | 64.23 | 5.14 | 30.63 | 92.59 |
| TT | 2.05 ** | 1.591 | 16.82 | 27.56 | 55.62 | 37.90 |
| CR | 3.576 ** | 1.09 | 28.26 | 18.18 | 53.56 | 60.85 |
| CaDe | 2.447 ** | 1.222 | 20.73 | 21.86 | 57.41 | 48.68 |
| TGV | 3.334 ** | 0.982 | 27.36 | 17.02 | 55.62 | 61.65 |
| BT | 12.341 ** | 2.288 ** | 51.53 | 20.16 | 28.30 | 71.88 |
| BC | 2.169 ** | 0.962 | 19.76 | 18.50 | 61.75 | 51.64 |
| LS | 2.203 ** | 0.81 | 20.61 | 15.99 | 63.40 | 56.31 |
| LAS | 3.348 ** | 0.946 | 27.62 | 16.47 | 55.91 | 62.65 |
| SL | 1.909 | 0.938 | 17.89 | 18.56 | 63.54 | 49.09 |
| LUSC | 1.348 | 1.338 | 12.31 | 25.80 | 61.89 | 32.29 |
| LLSC | 1.816 | 1.48 | 15.50 | 26.66 | 57.85 | 36.76 |
| PSS | 11.82 ** | 2.826 ** | 48.12 | 24.29 | 27.59 | 66.45 |
| FDD | 1.348 | 2.36 ** | 10.28 | 38.01 | 51.71 | 21.29 |
| FC | 2.316 ** | 1.33 | 19.46 | 23.59 | 56.95 | 45.20 |
| Mean | | | 27.28 | 21.49 | 51.23 | 54.15 |

** $p < 0.01$; phenotypic trait abbreviations are shown in Tables 2 and 3.

### 3.2. Variation Degree of Phenotypic Traits

The variation range and degree of 21 quantitative traits of 10 populations are reflected by the mean value, standard deviation, and coefficient of variation (CV) (Tables S1 and 5). The results show that there was abundant phenotypic variation in the quantitative traits of *E. angustifolia*.

The variation levels of different populations were different. The average CV of nine in 10 populations was more than 20%. The highest CV was found in the Qilihe population (30.58%), and the lowest was in the Ganzhou population (19.51%), but still close to 20%.

There were also significant differences in the variation levels of different traits. The CV of each trait ranged from 11.05% to 61.61%, with an average of 30.72%. Among them, the trait with the highest CV was under-branch height (UBH) (61.61%). At the same time, the CVs of the diameter at breast height (DBH) (60.87%), leaf area (LA) (46.93%),

branching angle (BA) (43.64%), and crown diameter (CrD) (41.64%) were also large, with rich polymorphism and variability. The flower diameter (FD) (11.05%) and calyx tube length (CTL) (12.04%) are the two traits with the smallest CVs and relative stability. In general, the CV of traits related to flowers and leaves was small, while the CV of traits related to trunk shape was large. The CV of the same trait was also significantly different among different populations.

**Table 5.** Coefficients of variation and Shannon–Wiener diversity indexes of phenotypic quantitative traits of *E. angustifolia*.

| Traits | Coefficient of Variation (CV) (%) | | | | | | | | | | | Diversity |
| | DH | SZ | LZ | GZ | YC | MQ | GL | QLH | YJ | LX | Total | |
| --- | --- | --- | --- | --- | --- | --- | --- | --- | --- | --- | --- | --- |
| TH | 37.92 | 28.24 | 27.65 | 10.47 | 16.57 | 46.47 | 16.52 | 33.06 | 36.66 | 43.43 | 37.12 | 1.9570 |
| CrD | 64.17 | 25.65 | 19.03 | 11.19 | 42.61 | 26.56 | 21.21 | 57.22 | 68.54 | 40.08 | 41.64 | 1.7890 |
| DBH | 60.76 | 21.28 | 26.64 | 15.09 | 25.92 | 58.15 | 70.13 | 95.17 | 75.98 | 76.90 | 60.87 | 1.9219 |
| UBH | 30.94 | 75.83 | 42.93 | 47.86 | 56.19 | 33.00 | 38.64 | 95.44 | 45.48 | 23.39 | 61.61 | 1.6802 |
| BA | 47.26 | 28.04 | 45.22 | 23.89 | 20.39 | 48.32 | 38.03 | 64.44 | 31.73 | 31.40 | 43.64 | 1.9106 |
| BN | 37.97 | 30.87 | 32.22 | 20.32 | 40.00 | 37.67 | 27.20 | 30.62 | 33.33 | 29.75 | 32.79 | 1.0927 |
| LL | 18.42 | 14.57 | 26.38 | 27.67 | 24.62 | 17.67 | 16.49 | 20.23 | 11.56 | 17.82 | 21.82 | 2.0415 |
| LW | 20.37 | 13.13 | 27.57 | 21.94 | 23.07 | 25.04 | 14.58 | 18.43 | 7.55 | 14.81 | 23.15 | 1.9679 |
| LTh | 14.64 | 12.35 | 19.63 | 17.03 | 9.25 | 19.81 | 16.80 | 14.48 | 5.26 | 3.54 | 17.93 | 2.0265 |
| PL | 23.42 | 22.38 | 23.97 | 24.64 | 24.77 | 20.67 | 18.45 | 16.62 | 7.67 | 10.40 | 23.06 | 1.9730 |
| LSI | 17.23 | 13.79 | 16.56 | 8.73 | 14.56 | 18.45 | 18.12 | 15.20 | 12.68 | 20.45 | 17.98 | 2.0085 |
| LA | 46.36 | 24.40 | 36.72 | 46.63 | 46.96 | 41.90 | 23.36 | 35.04 | 12.93 | 25.61 | 46.93 | 1.8967 |
| ABL | 26.11 | 40.24 | 25.52 | 23.31 | 20.75 | 21.86 | 13.83 | 23.23 | 18.26 | 30.32 | 31.65 | 2.0228 |
| NFC | 17.53 | 13.84 | 20.18 | 5.27 | 10.75 | 28.46 | 25.39 | 24.38 | 22.05 | 46.43 | 25.91 | 1.9858 |
| RFB | 24.17 | 25.19 | 34.13 | 30.23 | 20.08 | 27.03 | 16.87 | 15.80 | 15.78 | 24.87 | 26.66 | 2.0117 |
| FD | 13.16 | 7.47 | 9.09 | 7.33 | 18.57 | 8.86 | 8.06 | 5.27 | 9.12 | 4.68 | 11.05 | 2.0723 |
| CTL | 11.76 | 14.49 | 13.21 | 12.63 | 7.26 | 10.53 | 10.23 | 9.95 | 13.28 | 5.63 | 12.04 | 2.0556 |
| CTW | 13.10 | 10.03 | 10.18 | 10.83 | 14.85 | 23.55 | 9.63 | 13.35 | 16.75 | 9.73 | 19.71 | 1.9836 |
| FSL | 27.96 | 14.04 | 37.46 | 19.87 | 44.76 | 29.83 | 14.63 | 20.19 | 26.03 | 28.64 | 35.55 | 1.9125 |
| FCN | 37.40 | 47.13 | 34.00 | 24.83 | 24.83 | 31.44 | 28.41 | 34.08 | 34.55 | 35.35 | 36.52 | 0.9593 |
| FNAL | 0.00 | 0.00 | 0.00 | 0.00 | 0.00 | 0.00 | 0.00 | 0.00 | 0.00 | 59.88 | 17.49 | 0.2449 |
| Mean | 28.13 | 23.00 | 25.16 | 19.51 | 24.13 | 27.39 | 21.27 | 30.58 | 24.06 | 27.77 | 30.72 | 1.7864 |

Phenotypic trait abbreviations are shown in Table 2.

### 3.3. Diversity Index of Phenotypic Traits

It can be seen from Table 6 that there are different degrees of difference in the distribution frequency of 14 quality traits. Among them, the number of sample plants with a trunk type (TT) of grade 3 accounted for 80.0%, indicating that *E. angustifolia* was dominated by arbors and less distributed in shrubs within the scope of the investigation. The number of sample plants with 1-level crown roundness (CR) accounted for 55.6%, indicating that the overall plant shape of the sample plants was relatively disordered. The proportions of tree growth vigor (TGV) and flower density degree (FDD) of grade 3 reached 61.1% and 51.1%, respectively. More than half of the tested trees had a higher growth vigor and dense flowers, indicating that *E. angustifolia* could grow normally in various habitats. Branch thorns (BT) only existed in a small number of samples (34.4%). The dark brown and brownish green branches (BC) accounted for a relatively high proportion (42.2%, 32.2%), and the distribution of flower color (FC), leaf upper surface color (LUSC), leaf lower surface color (LLSC), and speckles on leaves (SL) was small at various levels. In addition, the petal spreading state (PSS) was mainly explanate (77.8%), the leaf shape (LS) was mostly lanceolate (48.9%), and the leaf apex shape (LAS) was mainly acuminate (62.2%).

According to the Shannon–Wiener diversity indexes of the 14 quality traits, the diversity of the different traits was different (Table 6). The diversity indexes ranged from 0.53 to 1.81, with an average of 1.0294. The diversity of leaf upper surface color (LUSC) was the highest (1.81). The diversity of petal spreading state (PSS) was the lowest (0.55). The

phenotypic diversity index of 21 quantitative traits ranged from 0.24 to 2.07 (Table 5), with an average of 1.7864. Among them, the diversity of the flower density degree (FDD) was the highest (2.0723). The diversity indexes of leaf length (LL), leaf thickness (LT), leaf shape index (LSI), annual branch length (ABL), ratio of flower cluster number to branch length (RFB), flower diameter (FD), and calyx tube length (CTL) also exceeded 2. The diversity of the floret number in axils of leaves (FNAL) was the lowest (0.2449).

In general, 21 quantitative traits had a high phenotypic diversity, and it was much higher than their quality trait diversity indexes, indicating that the phenotypic quantitative traits of *E. angustifolia* were relatively more abundant and the quality traits were more stable. From the overall results, the flower and leaf parts of *E. angustifolia* have a high degree of diversity.

**Table 6.** Frequency distribution of quality traits and Shannon–Wiener diversity indexes of *E. angustifolia*.

| Traits | Distribution Frequency of Each Grade | | | | | | | Diversity |
| --- | --- | --- | --- | --- | --- | --- | --- | --- |
| | 1 | 2 | 3 | 4 | 5 | 6 | 7 | |
| TT | 11.1 | 8.9 | 80.0 | – | – | – | – | 0.6378 |
| CR | 55.6 | 23.3 | 21.1 | – | – | – | – | 0.9945 |
| CaDe | 33.3 | 36.7 | 30.0 | – | – | – | – | 1.0953 |
| TGV | 8.9 | 30.0 | 61.1 | – | – | – | – | 0.8773 |
| BT | 65.6 | 34.4 | – | – | – | – | – | 0.6439 |
| BC | 5.6 | 32.2 | 20.0 | 42.2 | – | – | – | 1.2114 |
| LS | 26.7 | 6.7 | 12.2 | 48.9 | 5.6 | – | – | 1.3003 |
| LAS | 37.8 | 62.2 | – | – | – | – | – | 0.6630 |
| SL | 26.7 | 41.1 | 32.2 | – | – | – | – | 1.0828 |
| LUSC | 4.4 | 24.4 | 17.8 | 5.6 | 16.7 | 21.1 | 10.0 | 1.8076 |
| LLSC | 43.3 | 44.4 | 12.2 | – | – | – | – | 0.9797 |
| PSS | 77.8 | 22.2 | – | – | – | – | – | 0.5297 |
| FDD | 20.0 | 28.9 | 51.1 | – | – | – | – | 1.0236 |
| FC | 11.1 | 27.8 | 24.4 | 16.7 | 20.0 | – | – | 1.5648 |
| Mean | | | | | | | | 1.0294 |

Phenotypic trait abbreviations are shown in Table 3.

### 3.4. Correlations among Phenotypic Traits

Pearson's correlation coefficient was used to show the correlations between traits (Figure 3A, Table S2), and there were different degrees of correlation among 35 phenotypic traits of *E. angustifolia*. It was found that 84 pairs of traits were significantly correlated ($p < 0.01$) and 62 pairs of traits were significantly correlated ($p < 0.05$), including TH, CrD, DBH, and UBH; BA and BN; LL, LW, PL, LA, ABL, CTW, and FSL; ABL, NFC, CR, and CaDe; and LSI, LS, and LAS. At the same time, the following were also positive correlated with each other: between TGV and CR, CaDe; between TH and ABL, TT; between CrD and SL; between FNAL, and DBH, LW, NFC; between FDD and DBH, NFC, RFB, CaDe, TGV; between LUSC and LLSC; and between PSS and FD, TGV ($p < 0.01$). This shows that these morphological characteristics are mutually reinforcing and a common variation. At the same time, there were significant negative correlations ($p < 0.01$) between LTh and ABL, NFC, FNAL, CADE; between LSI and DBH, LW, FNAL; between RFB and LL, LW, PL, LA, ABL, FD, CTW, FSL; between TT and PSS; between CR and LUSC, LLSC; between TGV and DBH, UBH, BA; between BT and LW, LA, ABL, FNAL; and between SL and LUSC LLSC.

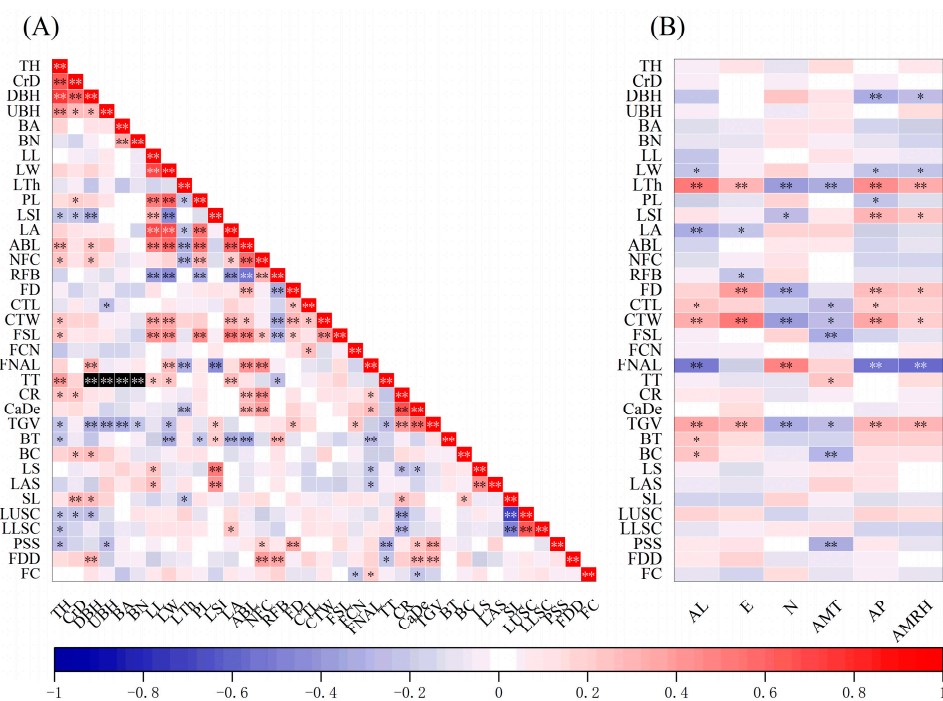

**Figure 3.** Correlation heat map of phenotypic traits of *E. angustifolia*. (**A**) Correlations among 35 phenotypic traits; (**B**) Correlations between phenotypic traits and geo-climatic factors. ** $p < 0.01$, * $p < 0.05$. Black zone: because at least one variable was constant, calculations could not be performed. Geo-climatic factors and phenotypic trait abbreviations are shown in Tables 1–3.

### 3.5. Correlation between Phenotypic Traits and Geo-Climatic Factors

Some phenotypic traits of *E. angustifolia* were strongly correlated with geographical and climatic factors (Figure 3B, Table S2). Among them, LTh, CTW, and TGV were positively correlated with altitude, longitude, annual precipitation, and annual mean relative humidity and negatively correlated with latitude and annual mean temperature. FD was positively correlated with longitude, annual precipitation, and annual mean relative humidity and negatively correlated with latitude. On the contrary, between FNAL and altitude, annual precipitation and annual mean relative humidity showed a significant negative correlation, and latitude showed a significant positive correlation. In addition, there was a significant or extremely significant positive correlation between CTL, BT, BC, and altitude, TT and annual average temperature, LSI, CTL, and annual precipitation, and LSI and annual mean relative humidity. At the same time, there were significant or extremely significant negative correlations between LA, LW, and altitude, LA, RFB, and longitude, LSI and latitude, CTL, FSL, BC, PSS, and annual mean temperature, DBH, LW, PL, and annual precipitation, and DBH, LW, and annual mean relative humidity. The above results show that in the phenotypes of *E. angustifolia*, the indexes related to leaf and flower parts were affected by geographical and climatic factors to a certain extent, among which precipitation and altitude had a greater influence on the phenotypic traits of *E. angustifolia* than other factors.

### 3.6. Principal Component Analysis of Phenotypic Traits

Principal component analysis was used to reduce the dimension of 35 phenotypic traits of *E. angustifolia*, and the top 11 principal components with the largest proportion were extracted. The eigenvalues were all greater than 1, and the cumulative contribution rate was 72.33%, indicating that these 11 principal components could objectively reflect the basic information of phenotypic traits of *E. angustifolia* (Table S3, Figure 4A). The contribution rate of PC1 was 15.32%, among which LL, LW, PL, LA, ABL, NFC, RFB, and FSL had higher loading degrees, which mainly reflected the characteristics related to the size of leaf and branch. The contribution rate of PC2 was 11.16%, among which TT, CR, CaDe, SL, LUSC,

LLSC, and FDD had higher loading degrees, which mainly reflected the characteristics of canopy growth and leaf color. The contribution rate of PC3 was 9.17%, among which TH, CrD, DBH, UBH, FD, TGV, and PSS had higher loading degrees, which mainly reflected the characteristics related to trunk shape and petal. The contribution rate of PC4 was 7.09%, and the loading degrees of LSI, LS, and LAS were higher, which mainly reflected the characteristics of leaf shape. The resultant seven principal components reflected the characteristics of leaf thickness, calyx size, and main branch, respectively.

The scatter plot constructed according to PC1 and PC2 shows the relationships between samples (Figure 4B). The geographical distribution of different populations had a certain effect on the phenotypic traits of *E. angustifolia*, and the effects on leaf size and branch traits (PC1) were greater than those of the crown and leaf color traits (PC2). The phenotypic traits of Dunhuang and Yongjing were relatively less affected by geographical factors, while the phenotypic traits of other populations were greatly affected by geographical factors. The samples of Linxia and Ganzhou were more concentrated in the scatter plot, while the sample scatter distribution of Suzhou and Gulang was the closest to consistency. Other population samples were more dispersed.

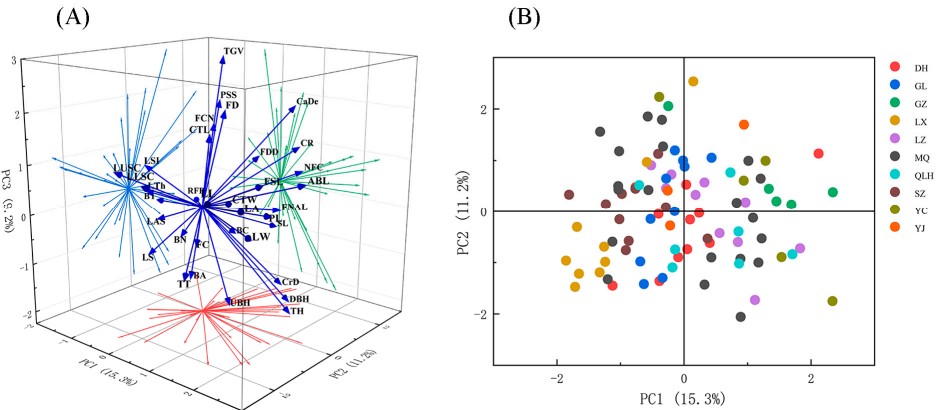

**Figure 4.** Principal component analysis results of phenotypic traits of *E. angustifolia*. (**A**) The load of phenotypic traits and its projections on the first three principal component axes. The length of the line represents the size of the load.; (**B**) Scatter plot of 90 samples based on PC1 and PC2. Phenotypic trait abbreviations are shown in Tables 2 and 3.

*3.7. Cluster Analysis of E. angustifolia Samples*

The Ward method was used for cluster analysis based on the squared Euclidean distance, and 90 samples were clustered into four clusters according to their morphological characteristics (Figure 5A, Table S4). Cluster A contains three subclasses, a total of 37 samples, most of which are from Dunhuang, Suzhou, and Yongjing. The flowers of cluster A are small and their branch color is light, but their flower color is the darkest. Cluster B contains six samples, all from Linxia, which are significantly different from other clusters. The characteristics of this cluster are that there is only one floret at each leaf axil, the leaves of the samples are thick, the branches are short, the number of flowers is few, and branch thorns are more abundant. At the same time, the flowers are longer, the petals are all explanate, the flower color and leaf lower surface color are the lightest, and speckles on leaves are less covered, so the leaf upper surface color is the deepest. Cluster C contains 18 samples in two subclasses, mainly from Gulang, Minqin, Linze, Suzhou, Dunhuang, and Yongchang. The samples of this cluster are all shrubs, with a low tree height and crown diameter, high crown roundness and canopy density, dense flowers, most petals curled outward, and dense speckles on the leaf surface, so their leaf color is the lightest. Cluster D contains 29 samples in two subclasses, mainly from Minqin, Ganzhou, Yongchang, Qilihe, and Linze. The tree height, crown diameter, diameter at breast height, and under-branch height of this cluster are higher than those of the other three clusters. There are fewer

branch thorns, longer branches, and larger flowers, a larger leaf size and petiole length, and the leaves are closer to oval-shaped.

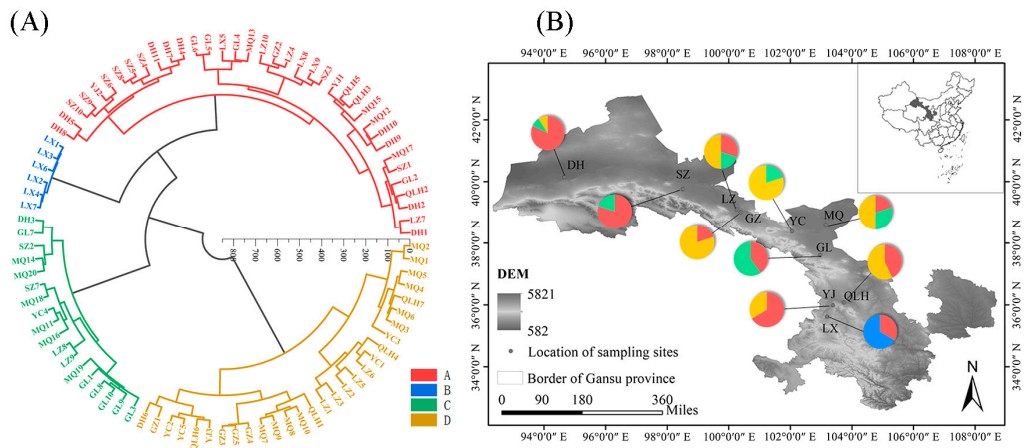

**Figure 5.** Ward clustering analysis results based on the squared Euclidean distance. (**A**) Cluster tree of 90 samples; (**B**) Geographical distribution of 4 groups of *E. angustifolia* in 10 populations. Phenotypic trait abbreviations are shown in Tables 2 and 3.

## 4. Discussion

In woody plants, especially in widespread species, complex environmental conditions, long-term geographical isolation, and natural selection are likely to cause intraspecific phenotypic variation, and the variation in phenotypic traits is affected by both genetic and environmental factors [33]. *E. angustifolia* is widely distributed in most areas of Gansu Province, especially in the counties and districts of the Hexi Corridor [34]. There are great differences in altitude, temperature, air pressure, annual precipitation, and soil. The changeable natural conditions and strong regional differences make some traits of *E. angustifolia* produce genetic differentiation among different populations.

In this study, the phenotypic variation of *E. angustifolia* was quite rich, and the differences in traits among different populations were extremely significant ($p < 0.01$), while the differences in most traits within the population were not significant. The phenotypic differentiation coefficient is the percentage of variation among populations in the total variation, reflecting the differentiation of species in traits. The higher the coefficient, the easier the population differentiation. Our study showed that the average phenotypic differentiation coefficient among *E. angustifolia* populations was 54.15%, indicating that the genetic differentiation degree of phenotypic traits of *E. angustifolia* was relatively high among populations. This may be due to the large differences in climatic and environmental conditions among the distribution areas of various populations investigated in this study. Moreover, *E. angustifolia* is mostly scattered in the suburbs and around villages within the natural distribution range. Individual natural populations are relatively intact, but most populations are seriously disturbed by human disturbances, resulting in changes in the natural habitat of *E. angustifolia*. In addition, the scattered distribution of discontinuous habitats and fewer samples hinder the gene flow among populations, resulting in population differentiation and phenotypic trait variation [35]. In our survey, we found that the floret number in axils of leaves of the other nine populations except the Linxia population was 3, while the floret number in most samples of the Linxia population was only 1, which is the reason for the high differentiation of this trait among populations. We infer that this situation may have been caused by differences in varieties or geographical isolation. However, due to the short length of the annual branch of the Linxia population and the relatively high ratio of flower cluster number to branch length (Table S1), the distribution of each flower cluster on the branches is relatively dense. The differentiation of flower density degree among all populations was low and relatively stable. The average variance component among populations was greater than that within populations, which further

indicated that the phenotypic traits of *E. angustifolia* in Gansu Province varied greatly, and the variation mainly came from populations. The phenotypic variation among populations was higher than that within populations, which was consistent with the results of analyses of *Pinus sylvestris* [36], *Lonicera caerulea* [37], and *Haloxylon ammodendron* [38]. This may be related to the geographical isolation of each population, which hinders the pollination of wind-pollinated plants and the gene exchange among populations and increases the differentiation among populations. The variation among populations reflects the difference between geographical isolation and reproductive isolation, which is an important manifestation of intraspecific diversity [39]. This high phenotypic variation made *E. angustifolia* have a good selection potential and laid a foundation for later genetic improvements.

The coefficient of variation is an important indicator of the degree of variation in statistics, which can reflect the discrete characteristics of a trait's value. The larger the value, the greater the dispersion of the trait [40]. The variation levels of 10 populations of *E. angustifolia* in this study were quite different. The maximum and minimum coefficients of variation were found in the Qilihe population and Ganzhou population, respectively. The reason for the large variation in the Qilihe population may be due to the different distribution environments of the seven samples in the population. Some of them grow near an abandoned train track, and other parts grow on a construction site, resulting in significant phenotypic differences. The reason for the small variation in the Ganzhou population may be that the *E. angustifolia* is mostly distributed in wetland parks or cultivated as a street tree. After regular irrigation, pruning, and maintenance, the growth of plants is affected by human intervention. In addition, the coefficient of variation among different phenotypic traits was also quite different. The coefficient of variation of 21 quantitative traits ranged from 11.05% to 61.61%. Among them, the variation in under-branch height, diameter at breast height, branch angle, and crown diameter was greater, which may be due to the fact that trunk traits are more susceptible to tree age than other parts. This is similar to the results of the analysis of *Quercus gilva* by Qin et al. [35]. The traits with a high coefficient of variation have a strong adaptability to their environment and increase their selection opportunities among materials. The traits with low coefficients of variation have a homology and stability among materials [23]. In addition, the coefficient of variation of the leaf area of *E. angustifolia* was also higher than other traits, which could be explained by the fact that leaf area plays an important role in effective light capture, water balance, and temperature regulation in different habitats [41,42].

Within the scope of our investigation, *E. angustifolia* is dominated by arbors. The reason for the lesser distribution of shrubs is on the one hand due to historical development and artificial selection. In recent decades, *E. angustifolia* has been used as a windbreak and in sand-fixing forests, economic forests, and as a garden solitary tree in Northwest China, which makes people plant and use arbors more widely than shrubs [17,43–45]. On the other hand, because most of the sample trees selected in this study are more than 10 years old, the older trees are mostly arbors, and the age of shrubs is generally smaller. The proportion of crown clutter is higher, more than half of the surveyed sample plants. Most of the sample plants selected in our survey are individual trees growing independently in a wild natural environment rather than rows or pieces of afforestation trees. Therefore, the growth of crowns and branches is hardly blocked and hindered by surrounding trees. At the same time, it is less affected by human intervention, so the overall crown presents a more natural cluttered state [46]. Dark brown and brownish green branches accounted for a relatively high proportion of their color, while the distribution of some traits of petals and leaves at all levels was small, which may be due to the fact that the branches we investigated were perennial old branches. After a long period of growth and adaptation to the environment, several main colors and states have formed. Florets and leaves are new parts of the year, and because climate change makes them more prone to unstable multiple manifestations, they are more diverse and less distributed at all levels.

As an important index to measure the richness of plant traits, the Shannon–Wiener diversity index is widely used in the evaluation of genetic resources and the analysis of

phenotypic diversity [32,47]. According to the study of phenotypic diversity of *Prunus mume* by Wu et al., the Shannon index reaches 1, which is a high degree of diversity [48]. In this study, the average Shannon index of the quality and quantitative traits of the tested *E. angustifolia* was greater than 1, and some traits related to flowers and leaves showed a high level of diversity. In the future, the index could be used as a priority for the analysis of phenotypic diversity of *E. angustifolia*. The diversity of the petal spreading state is low. In our survey, it was found that the four petals of each flower of *E. angustifolia* generally showed two different states, which were explanate or curled outward (Figure 2A). We speculate that this difference may be due to different varieties, or may be related to the time of flowering. In addition, the outward curling petals are more expanded than the explanate petals. Because the flowers of the deciduous group of *Elaeagnus* are all hermaphroditic flowers, they are divided into two types: fertile flowers and abortive flowers. The perianth lobes of abortive flowers expand to a small extent, and the perianth lobes of fertile flowers expand to a large extent, which creates very favorable conditions for pollination [49,50]. This study also found that the diversity index of quantitative traits was much higher than that of qualitative traits. Li et al. and Szamosi et al. also obtained consistent results for the morphological traits of blueberry and *Cucumis melo* germplasm resources [51,52]. The reason for this may be that quantitative traits are greatly affected by germplasm resource types, environmental conditions, genotypes, and other factors, while qualitative traits change only slightly and are relatively more stable. In addition, from the overall analysis results of quantitative traits, the variation in the Shannon index and coefficient of variation (CV) is not consistent. For example, the Shannon index of the flower diameter is the largest (2.0723), while the CV is the smallest (11.05%). The Shannon indexes of the calyx tube length, leaf thickness, and leaf shape are also greater than 2, but the CV is far lower than 20%. The larger the coefficient of variation, the greater the degree of variation of traits; the larger the diversity index, the richer the trait diversity. However, there is no correlation between the two. This study is also consistent with the results of He et al. and Fu et al.'s phenotypic studies on *Dendrobium nobile* and *Salvia splendens* [53,54].

The correlation of traits can indirectly achieve the effect of selecting another trait through the selection of one trait, which can improve the selection efficiency of excellent plant varieties and accelerate the process of breeding [55]. The correlation analysis of phenotypic traits of *E. angustifolia* in Gansu Province showed that the correlation between most traits was significant or extremely significant. This is consistent with the previous research results of Khadivi on *E. angustifolia* [56]. Correlation analysis has a certain convenience and guidance for the field collection and breeding of *E. angustifolia*. Because most traits are closely related, researchers can roughly infer the growth trends of other traits related to one trait, which can reduce the blindness of investigation and research to a certain extent [57]. However, the linkage mechanism between these traits is still unclear and may need to be explored from biochemical and molecular perspectives.

Different species have different geographical variation patterns due to their different adaptability and sensitivity to the environment [58]. From this study, it was found that the traits related to the trunk and branch of *E. angustifolia* were not significantly affected by environmental factors, the leaf traits were mainly affected by altitude and rainfall, and the flower traits were synergistically affected by various geographical and climatic factors. In the appropriate range, under the conditions of high altitude, high longitude, rainy, low latitude, and low temperature, *E. angustifolia* has higher vitality, thicker leaves, and larger and longer flowers. The size of flowers in hot, dry, and low-altitude areas is smaller. This difference can be explained on the basis of pollinator-mediated selection on the one hand [59–61], and on the other hand, because reducing corollas is beneficial for plant growth under harsh conditions [62–64]. Relatively wet and fertile soil can make plants produce larger flowers [65]. At the same time, with a decrease in altitude, longitude, and precipitation, the size of *E. angustifolia* leaves showed an increasing trend. Previous studies have also found that the leaf area of many plants decreases with increasing altitude, while the leaf thickness increases with increasing altitude [66,67]. This is consistent with

our findings. Plants can adapt to environmental changes accompanied by a high altitude by reducing their leaf area and increasing their leaf thickness, mesophyll tissue thickness, and stomatal density [68].

In a study of *E. angustifolia* in Northwest China, Wu pointed out that an altitude of 1000–1700 m is generally suitable for its growth. It is difficult for *E. angustifolia* to survive in dark and humid mountainous areas or alpine areas with an altitude of more than 2000 m. Our study also further confirms this conclusion; among the 10 populations, the southernmost Linxia County is in the transitional zone between temperate semi-humid and alpine humid areas. The altitude and annual precipitation exceed 2000 m and 500 mm, respectively. *E. angustifolia* is less distributed in this area, and plants generally show a small diameter at breast height, low canopy density, narrow leaves, short branches, and few flowers. Only one floret grows at each leaf axil. The growth of *E. angustifolia* may be affected by high altitudes and cold and humid climates. We also found that the growth of *E. angustifolia* in other areas with lower latitudes and higher altitudes is very rare.

Principal component analysis can reduce the dimension of a dataset, maintain the characteristics of the largest variance contribution in the dataset, and provide the main information on the comprehensive performance of quantitative traits [54]. In this study, the first 11 principal components were extracted from the phenotypic traits of *E. angustifolia* by principal component analysis, and the cumulative contribution rate reached 72.33%. Among the traits related to the plant shape, branch, leaf, and flower of *E. angustifolia*, the primary and key traits affecting the phenotypic diversity are the traits related to leaf size and branch length, which is similar to the phenotypic analysis results of *E. angustifolia* in Zhangye City by Zeng et al. In future investigations and studies, these traits could be used as key observational traits. Principal component analysis also further showed that the phenotype of *E. angustifolia* was greatly affected by the geographical distribution of different populations. Most of the phenotypic traits of the two populations in Linxia and Ganzhou were stable, while the phenotypes of the two populations in Suzhou and Gulang were closer. The traits of other populations were more abundant and the degree of variation was higher, which may be due to genetic or environmental effects [69].

The purpose of clustering analysis is to classify individuals according to their characteristics. Individuals in a cluster are highly similar to each other, and the inhomogeneity and differences among clusters are the largest [23]. According to the clustering results of *E. angustifolia*, the geographical distribution of four clusters in 10 populations was obtained (Figure 5B). The 90 samples were not completely clustered according to geographical distance. The distribution of the four clusters was overlapped, but the distribution of the same cluster had a certain degree of similarity, and the variation in phenotypic traits among populations was discontinuous. This shows that geographical distance does not play a key role in the phenotypic differentiation of the studied *E. angustifolia* population [70]. This may be due to the complex and diverse geographical environment and climate differences in Gansu Province. In addition to the specific phenotypic characteristics, the population also has a specific ecological amplitude and distribution range [71]. The random effects of various specific genotypes or environmental factors often lead to differences in traits between individuals [72]. Among the seven samples in Linxia County, the southernmost part of the survey area, six samples were clustered into cluster B alone, which was most significantly affected by geographical and climatic differences. The distribution of cluster C of shrubby *E. angustifolia* is relatively concentrated. In the process of the field investigation, it was found that shrubby *E. angustifolia* is more distributed in Minqin County, Gulang County and surrounding counties than in other regions, mostly in the forms of a shrub forest belt or small windbreak forest. The distribution of cluster A *E. angustifolia* is relatively loose, from the westernmost Dunhuang to the easternmost Qilihe area, indicating that cluster A may be a transitional group, and the characteristics of each trait in the group are not obvious. The cluster D *E. angustifolia* is almost entirely distributed in the region of the latitude 36–39° N. In the future, the directional breeding of excellent varieties of *E. angustifolia* could be carried out based on different uses.

## 5. Conclusions

The phenotypic variation in different populations of *E. angustifolia* is rich, and the variation mainly comes from among populations. The phenotypic traits related to plant shape had a large variation, and the phenotypic traits related to flowers and leaves had a small variation. The phenotypic quantitative traits of *E. angustifolia* were relatively more abundant, the qualitative traits were more stable, and the diversity of flowers and leaves was higher. Within a certain range, with an increase in altitude, longitude, and rainfall, the phenotype of *E. angustifolia* showed a trend of higher tree vigor, larger and longer flowers, smaller leaves, and thicker leaves. However, excessive altitude and precipitation make it unable to survive and grow poorly. As a topical original study of this species in Gansu Province, the results of our study on the phenotypic trait variation in *E. angustifolia* will provide a basis for improving the efficiency of population selection, provide a reference for selecting and establishing gene conservation areas and seed collection areas, and is of great significance for accelerating the selection and breeding. In the future, the corresponding varieties of *E. angustifolia* should be selected for different purposes according to their geographical distribution and utilization direction. For example, the natural growth of individual *E. angustifolia* plants is arborous when growing independently, and they often form shrubs or small windbreaks when growing in patches. Using the superior plant shape and crown of cluster D in this study and its characteristics of less thorns and larger flowers and leaves, it can be considered as an isolated flower tree species and can also be planted as a street tree in the city. Cluster C can be used as hedge or sparse forest. Its leaf surface scales are dense, and its leaf color is closer to silvery white. It can be compared with green leaf plants in its garden configuration, which is more prominent in landscape. In short, for some varieties with bright and full fruits, no or few spines on branches and long flowering periods should be directionally cultivated to gradually update degraded varieties, to cultivate a set of ornamental and ecological functions as one of the sustainable development of the fine varieties. In addition, the floral scent of *E. angustifolia* can also be used as a characteristic and function of smell in landscapes. However, at present, the judgment of floral scent in our field investigation can only rely on human perception, lacking objectivity and rigor. Therefore, it has not been added to the analysis of phenotypic diversity. In the subsequent research, more professional instruments will be used to determine the degree of floral scent, to understand the phenotypic diversity and ornamental value of *E. angustifolia* more comprehensively.

**Supplementary Materials:** The following supporting information can be downloaded at: https://www.mdpi.com/article/10.3390/f14061143/s1, Table S1: The mean value, standard deviation, and Duncan's multiple comparison of 21 quantitative traits of *Elaeagnus angustifolia* from different populations. Table S2: Pearson correlation coefficients between each trait and Pearson correlation coefficients between traits and geographical climatic factors. Table S3: Principal component vectors, eigenvalues, contribution rates, and cumulative contribution rates of the first 11 principal components based on tree means for the 35 traits. Table S4: Descriptive statistics of phenotypic traits of four clusters.

**Author Contributions:** Methodology, software, and writing—original draft preparation, R.S.; resources, conceptualization, and writing—review and editing, Z.Z.; investigation, N.S. and R.S.; formal analysis and validation, Y.W., Y.M. and Y.L.; data curation, J.D., X.X. and T.L. All authors have read and agreed to the published version of the manuscript.

**Funding:** This research was funded by the Forestry and Grassland Science and Technology Innovation and International Cooperation Project of Gansu Province (kjcx2021004) (Institutions providing this funding: Gansu Forestry and Grassland Administration), Youth Science and Technology Fund Program of Gansu Province (20JR5RA007) (Institutions providing this funding: Science and Technology Department of Gansu Province), and Research Startup Fund of Gansu Agricultural University (GAU-KYQD-2020-5) (Institutions providing this funding: Gansu Agricultural University).

**Data Availability Statement:** Not applicable.

**Acknowledgments:** We would like to express our gratitude to the local forestry managers and the masses for all kinds of support and help during our investigation and sampling process. We also offer our great thanks to Jia Wei and Zizhen Li for their guidance and advice on our writing.

**Conflicts of Interest:** The authors declare no conflict of interest.

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
