# Peer review of "Phenotypic Diversity Analysis in Elaeagnus angustifolia Populations in Gansu Province, China"

_forests, doi:10.3390/f14061143_

Round 1

Reviewer 1 Report

Phenotypic Differences among and within Populations

1.      Line 220-222: The variance analysis of E. angustifolia phenotypic traits showed that 29 traits were significantly different among populations (p < 0.01), and only 9 traits were significantly different within populations (p < 0.01). What is the possible reason of it? Please support your result with the help of available literature.

2.      Line 227-229: The phenotypic differentiation coefficients of each trait among populations ranged from 21.29% to 92.59%, with an average of 54.15%. What is the possible reason of it? Please support your result with the help of available literature.

3.      Line 230-232: It shows that the florets number in axils of leaves is more differentiated among populations, while the flower density degree is less differen tiated and relatively stable. What is the possible reason of it? Please support your result with the help of available literature.

Variation Degree of Phenotypic Traits

4.      Line 242-245: The average CV of 9 populations in 10 populations was more than 20%. The highest CV was the Qilihe population (30.58%), and the lowest was the Ganzhou population (19.51%), but still close to 20%. What is the possible reason of it? Please support your result with the help of available literature.

5.      Line 252-254: In general, the CV of traits related to flowers and leaves was small, while the CV of traits related to trunk shape was large. What is the possible reason of it? Please support your result with the help of available literature.

Diversity Tndex of Phenotypic Traits

6.      Line 266-268: Among them, the number of sample plants with trunk type (TT) of grade 3 accounted for 80.0%, indicating that E. angustifolia was dominated by arbors and less distributed in shrubs within the scope of the investigation. What is the possible reason of it? Please support your result with the help of available literature.

7.      Line 268-270: The number of sample plants with 1-level crown roundness (CR) accounted for 55.6%, indicating that the overall plant shape of the sample plants was relatively disordered. What is the possible reason of it? Please support your result with the help of available literature.

8.      Line 375-278: The dark brown and brownish green branches (BC) accounted for a relatively high proportion (42.2%, 32.2%), and the distribution of flower color (FC), leaf upper surface color (LUSC), leaf lower surface color (LLSC), and Speckles on leaves (SL) was small at various levels. What is the possible reason of it? Please support your result with the help of available literature.

9.      Line 284: The diversity of petal spreading state (PSS) was the lowest (0.55). What is the possible reason of it? Please support your result with the help of available literature.

Conclusion

10.  Please rewrite conclusion which includes highlights of your results.

11.  What you recommend for future?

12.  What are the benefits of this study?

Minor revisions are needed for spelling and punctuation mistakes.

Reviewer 2 Report

Dear author: please consider my suggestions as ideas to improve your manuscript for this submission and further ones:

Abstract: is the length of the abstract (number of words) in accordance with the journal author guidelines?

Figure 1: I would increase the size of the map, maybe up to both side margins.

Line 157 (table above): What it is the quantitative “flower color number”?

Line 28: space between analysis and PCA

Line 41: although “stable” wants to say in your introduction, I understand that morphological characters are quite variable, so I would delete “stable” and leave only “diverse”.

Line 50: I would add that the study of traits is also used to define new varieties to be protected. For example, in Europe, there exist specific guidelines for crops (i.e. apple.

Line 55-56. Why are those species mentioned and not others? Please indicate that they are examples and maintain the recent references.

Line 67-75: why this species?

Line 141: hoy many years of samplings were done in this research? It is expected to use several years of measurements in morphological analyses in order to reduce the environmental effect, at least, in table 2, in LL, LW, LTh, PL, LSI, LA, ABL and so on; and in Table 3 in LS and so on.

Lines 207, 210-211, 226, etc: Please review the italics for the species in your manuscript.

Line 209: Principal Component Analysis is only with quantitative variables, please substitute the name by Principal Coordinate Analysis (PCoA) here and in the whole text, where necessary.

Line 264: Index instead of Tndex

Line 486. Is Wu a citation? There may lack the year of such reference

Discussion: are there any molecular study for your species? Please refer the study of uzun

https://ijans.org/index.php/ijans/article/view/406/395

https://plantmethods.biomedcentral.com/articles/10.1186/s13007-022-00915-w

Reviewer 3 Report

Dear Authors

Enclosed is minor comments for the forests-2392446- Phenotypic Diversity Analysis of Different Populations of Elaeagnus angustifolia L. in Gansu Province, China

Title:  It could be re-considered such as “Phenotypic Diversity Analysis in Elaeagnus angustifolia Populations in Gansu Province, China”

Abstract: It covers sufficiently and detail. 

Key words: They are suiteable.

Introduction: It is given by related references. Aim/s of the paper are given clearly at end of this part.

Materials and Methods: sub-title 2.3. Statistical Analysis M&M should be revised for better understable for readers such as “phenotypic differentiation coefficient”.

Results: Results are related to paper’s findings. They are supported by Table and Figures. However, subtitles of Results should be re-considered.

Discussion: Results of the paper are discussed by published papers.

Conclusions: They are extracted results of the paper.

References, Table and Figures: All references, table and figures should be kept in the paper. However, titles of Tables 4 and 5 re-considered.

1. The paper addressed variation/differentiation of Elaeagnus angustifolia in Gansu Province of China. Estimation of the variation is one of the main stages of plant breeding especially selection purposes.  

2. The topic original and relevant in the field for the province. I could be contributive in future studies in the species
.

3.
Results of the study could be contributive for selection and establishment of gene conservation areas, seed collection areas and other purposes.

4.
Materials and Methods: It is sufficient and detail However, sub-title 2.3. Statistical Analysis M&M should be revised.

5. The conclusions
are extracted from results of the paper.

6.
References are appropriate

7.
All table and figures are related to the paper. They should be kept in the paper. However, titles of Tables 4 and 5 re-considered.

Minor editing of English language required.

Round 2

Reviewer 2 Report

Dear author, thank you very much for your fast and clarifying answers.

I would like to comment the following comment and answer:

Comments 9: Line 141: hoy many years of samplings were done in this research? It is expected to use several years of measurements in morphological analyses in order to reduce the environmental effect, at least, in table 2, in LL, LW, LTh, PL, LSI, LA, ABL and so on; and in Table 3 in LS and so on.

Response: Thanks for your advice. This study conducted two years of sampling in 2021 and 2022, but it is regrettable that the sampling in 2021 was affected by the COVID-19 epidemic. Due to the inconvenience of travel caused by home policies and prevention and control work, we failed to collect complete data. In the future, our team will continue to conduct more surveys and sampling of E. angustifolia populations in our province to avoid the impact of climate change, make the research results more representative, and lay a better foundation for the collection of E. angustifolia germplasm resources and breeding.

The COVID-19 affected many scientific samplings and there are not many "jurisprudence" about how to takle their inconvenences when publishing partial data.

In my opinion, it would be adecquate to indicate in your M&M section that there exists some surveys and samplings of 2021 affected by COVID-19 pandemic. I will propose that opinion to the editor. 

Best regards,
